# Patterns of European Ant Communities Reveal a Functionally Coherent Holarctic Fauna

**Benjamin D. Hoffmann** [1,2,*] and **Alan N. Andersen** [2]

1 CSIRO, Tropical Ecosystems Research Centre, PMB 44, Winnellie, NT 0822, Australia
2 Research Institute for the Environment and Livelihoods, Charles Darwin University, Darwin, NT 0909, Australia
* Correspondence: ben.hoffmann@csiro.au

**Abstract:** Here we examine the extent to which European patterns of ant diversity and functional composition conform with those documented in North America. Following protocols previously used in North America, ant species distribution and behavioural dominance were quantified at fifteen sites on two environmental gradients, one following elevation (140–1830 m) in France and the other tree cover (0–95%) in Denmark. Pitfall traps were used to assess species distributions, and behaviour at tuna baits was used to inform behavioural dominance. We specifically test three predictions based on North American patterns: (1) Species richness and overall levels of behavioural dominance will decline with increasing thermal stress. (2) Geographic patterns of key taxa in Europe will be consistent with those in North America. (3) Behavioural dominance of European taxa will be consistent with related taxa in North America. We then use our results to classify the European ant fauna into functional groups, as had been done for North American ants. Based on these functional groups, we analyse distributional patterns along our gradients and re-analyse ant community data from published studies to provide a more comprehensive understanding of the structure of European ant communities. Distributional and behavioural predictions of the European ant taxa were consistent with those in North America. Geographical patterns of functional-group composition were very similar to those previously recorded for North America, varying systematically and predictably along the environmental gradients. Our findings indicate that there is a functionally coherent ant fauna throughout the Holarctic.

**Keywords:** ants; biogeography; community structure; community composition; dominance; environmental gradient





## 1. Introduction

Global comparisons of biotic composition and organisation represent a major challenge for community ecology because of taxonomic dissimilarity between distant regions. One solution is to classify species into functional groups that have universal currency [1–3], allowing for comparative analyses of communities, regardless of where they occur.

Comparative analyses are particularly challenging for faunal communities, where there are no simple analogues to the life forms that form the basis of vegetation mapping and analysis. A notable exception is for ants, where global-scale functional groups operating at the genus or subgenus level have been identified that parallel plant life forms in relation to environmental stress (factors reducing productivity) and disturbance (factors removing biomass) [4] (Supplementary Material S1). This has allowed for comparisons of community structure and function at multiple spatial scales spanning diverse bioregions [5–7], including comparative analyses of ant community responses to disturbance in the context of using ants as bio-indicators of ecological change [8–10].

The European ant fauna is among the most extensively studied taxonomically, but there have been surprisingly few European studies describing the structure of entire ant

communities. Most community-level studies focus on particular aspects, such as competitive interactions among the most common species e.g., [11–15], or have limited samples only (e.g., using only baits, which are biased towards recruit-foraging and aggressive species [16–18]). Similarly, although there have been macroecological studies of ant activity, richness and functional traits [19–22], there has been little attempt to document biogeographical patterns of community composition [23–25], which can be considered either as studies of sites separated widely on a biogeographic scale or studies that incorporate environmental gradients that otherwise emulate largely differing environmental conditions that can be encountered at a biogeographic scale (i.e., desert to a rainforest).

The European ant fauna is very similar taxonomically to that of temperate North America, where almost all European genera and common subgenera also occur. North America supports a richer ant fauna overall, with approximately 1000 species [26] compared with 640 in Europe [27], but North America includes extensive arid and subtropical regions that are very limited or not represented at all in Europe. Patterns of ant community composition along environmental gradients in Europe might, therefore, be expected to be very similar to those in temperate North America. In a very broad sense, this is already known to be true based on the taxonomic composition of regional faunas. For example, in North America, species of behaviourally dominant dolichoderinae (species of *Liometopum* and *Dorymyrmex*, along with the introduced *Linepithema humile*) are restricted to warmer regions, and in Europe this group (represented primarily by species of the *nigerrimum* group of *Tapinoma* and *L. humile*) is similarly distributed, occurring predominantly in the Mediterranean region. In cool-temperate regions of North America, mound-building species of *Formica* (*rufa* and *exsecta* groups) are the leading behaviourally dominant ants, with opportunist species of the *fusca* group of *Formica* being behaviourally submissive [5]; this is also the case in Europe [12]. However, the extent to which the functional composition of the European ant fauna is more broadly consistent with that in North America remains unknown.

Ant compositional patterns across gradients of rainfall, temperature and latitude have been documented for North America [5] based on functional groups in relation to environmental stress and disturbance that were originally developed in Australia and operating at the genus and subgenus level [4]. This scheme has since been widely applied in local studies throughout the world, including in Europe [7,28–31]. However, it has not been used to investigate biogeographic patterns of community composition across Europe.

Here we describe biogeographical patterns of ant species richness, behavioural dominance and functional-group composition in Europe using new data from environmental gradients of thermal stress, following the protocols used in North America [5], complemented by re-analyses of a range of data from published European studies. Temperature is the major factor driving biogeographic variation in ant diversity and composition [32,33], and this incorporates the level of direct insolation, mediated through vegetation cover, as well as ambient temperature [4]. Thermal stress, therefore, refers to temperature at the foraging surface and is determined by the interaction between ambient temperature and vegetation cover. We specifically test three predictions based on North American patterns (Supplementary Material S1): (1) Species richness and overall levels of behavioural dominance will decline with increasing thermal stress. (2) Geographic patterns of key taxa in Europe will be consistent with those in North America. Specifically, species of *Lasius* and the *Formica sanguinea*, *rufa* and *exsecta* species groups will be the most abundant Cold-climate Specialists, occurring primarily at high latitudes or high elevations at lower latitudes; Generalised Myrmicinae (species of *Crematogaster* and *Pheidole*) will show the reverse pattern, occurring primarily at low elevations and latitudes, and Opportunists (species of *Myrmica*, *Tetramorium* and the *Formica fusca* species group) will be widely distributed across environmental gradients. (3) Behavioural dominance of European taxa will be consistent with related taxa in North America. Cold-climate species of the *Formica rufa* and *exsecta* species groups will display the highest behavioural dominance; the Generalised Myrmicinea *Pheidole* and *Crematogaster* will display moderate levels of dominance and Opportunists the lowest.

## 2. Materials and Methods

### 2.1. Study Sites

Ants were sampled at 15 sites arranged along two environmental gradients of thermal stress that were selected to represent the major biomes and, therefore, ant taxa of continental Europe (Supplementary Materials S2–S5). Our aim was to document major patterns of functional composition, rather than a comprehensive sampling of European ant species. The first gradient contained eight sites spanning an elevational range in southern France (the low latitude sites), extending from Masif du Cap Canaille (43°10′ N 5°35′ E) at sea level near Marseille to Mont Saint-Guillaume near Embrun in the French Alps (1830 m asl; 44°35′ N 6°27′ E), spanning a distance of approximately 150 km. The gradient was representative of the vegetation structural range of southern Europe, from low shrublands to tall coniferous forest and supports all the major ant genera of central and southern Europe [34] (J Orgeas, pers. comm., 2010). All sites were separated by more than 500 m.

The second gradient contained seven sites within Mols Bjerge National Park (56°13′ N 10°34′ E) in Denmark (the high latitude sites). The gradient spanned the vegetation structural range of northern Europe, from grassland to tall, closed Beech (*Fagus sylvatica*) forest. This study region supports nearly 60% (about 40 out of 70) of all ant species occurring in northern Europe and includes all the major genera [35] (M.G. Nielsen, pers. comm., 2010). Sites were separated between approximately 200–500 m.

### 2.2. Sampling

To best allow a cross-continental comparison, field sampling followed the exact methodology used previously in North America [5]. It was conducted during sunny and warm weather in August (European summer) 2010, corresponding with the highest levels of ant activity. Pitfall trap catches were used to compare the relative forager abundances of species at each site. At each site, 15 pitfall traps were operated in a $5 \times 3$ grid with 10 m spacing for 48 h. Traps were 6.5 cm-diameter plastic cups, partly filled with ethylene glycol as a preservative. Specimens were sorted to species level and identified using regional keys and taxonomic revisions [34–36]. Vouchers for all species are lodged at the CSIRO Tropical Ecosystems Research Centre in Darwin, Australia.

Counts of ants at tuna baits were used to quantify the relative behavioural dominance of species at each site. Following the completion of pitfall trapping, a teaspoon of canned tuna in oil was placed at each trap point, and the abundance of each species at the baits was recorded after 5, 15, 30 and 60 min. An ant was deemed to be at a bait if it was observed feeding from it within a 10-s observation period. All baiting was conducted during mild to warm daytime conditions. Voucher specimens were collected from baits for later identification in a laboratory. Opportunistic hand collections were also made at each site to provide additional records of species occurrences.

Environmental data were collected from 1 m quadrats placed centrally over each of the 15 sample locations at each site. The percentage of each ground cover classification was visually estimated; litter depth was measured with a ruler; and tree canopy cover was measured using a hand-held spherical densiometer. These data were averaged for each site. A single maximum tree/shrub height was estimated for each site. Note that the environmental data was only for site description purposes, not for further analysis with ant data.

### 2.3. Analysis

Species collected from pitfall traps, baits and hand collections were combined to document species occurrences at each site. To investigate sampling completeness, species site occurrences from the 15 pitfall traps and the four baiting times from the 15 baits (n = 75 samples) were used to generate rarefaction curves using EstimateS [37]. Species abundances in pitfall traps and at baits were scored according to a 7-point scale following Andersen (1997a): 1 = 1 ant; 2 = 2–5 ants; 3 = 6–10 ants; 4 = 11–20 ants; 5 = 21–50 ants; 6 = 51–100 ants; and 7 > 100 ants. Species abundances per pitfall trap sample/baiting time

were calculated as the sums of abundance scores at individual traps or baits (i.e., maximum of 105 per sample in each case).

We used abundances at baits to characterise relative behavioural dominance based on multiple forms of evidence. Behaviourally dominant species were defined as those whose: (1) abundances at baits increased rapidly to the highest abundance scores; (2) were able to maintain those high abundance scores after 30 and 60 min, resulting in highly right-skewed frequency distributions; (3) relative abundances at baits were notably higher than in pitfall traps [5]. Note that statistics are not needed to make these three determinations, nor are statistics possible for these assessments without site-level replication.

*2.4. Functional Group Classification and Re-Analysis of Previous Studies*

We classified all European ant taxa into the functional groups established by [4] using data reported here, other distributional records [38], and assignments for North American taxa [5]. Functional group classifications were used to characterise compositional change along our two European bioclimatic gradients, based on pitfall-trap data. The classifications were then used to re-analyse previously published ant community data from other European sites sampled using pitfall traps to test the generality of our findings. The data used for these analyses were: a grassland site in Spain [39]; a heathland site in south west France (summer 1992 sample [40]); two forest sites combined in Spain [39]; shrubland sites F1 and F2 of the present study combined; three woodland sites combined in Italy [7]; woodland sites F3–F6 of the present study combined; forest sites F7–F10 of the present study combined; woodland sites D2 and D3 of the present study combined; forest sites D5 and D7 of the present study combined; and four forest sites combined in Finland (control sites [41]) (Supplementary Material S6). Functional group profiles were graphed, and sites were ordinated by nonmetric multidimensional scaling (nMDS) using Primer 6 [42]. The similarity matrix was constructed using a Bray–Curtis association index based on the percentage contribution data of each functional group at each site or site combination.

## 3. Results

*3.1. The Fauna*

A total of 34 species from 13 genera were recorded at the French sites, and 15 species from four genera in Denmark (Supplementary Material S7). Seven species were common to both countries. Pitfall trapping sampled the great majority (40 out of 42) of species collected. The most speciose genera were *Formica* (9 species), *Myrmica* (6), *Camponotus* (6) and *Lasius* (5). In France, the most abundant species in shrubland/woodland sites were *Pheidole pallidula*, *Lasius emarginatus*, *Plagiolepis pygmaea* and *Camponotus cruentatus*, and at forest sites *Formica aquilonia*, *Myrmica ruginodis*, *M. lobicornis*, *C. herculeanus* and *F. lemani*. In Denmark, the most abundant species in grassland/woodland sites were *M. sabuleti, M. ruginodis, F. fusca* and *F. rufa*, and in forest sites *F. rufa* and *M. ruginodis* (Supplementary Material S7).

Species rank-abundance curves from both pitfall catches and bait counts showed pronounced numerical dominance by a small number of species at each site (Supplementary Material S8). Indeed, the most abundant species at each site contributed on average 83% to total abundance in pitfall traps and 98% at baits. Species accumulation curves indicate that almost all species were sampled at Danish sites and the great majority of species at French sites (Supplementary Material S9). Site species richness in France ranged from 3 (site F6) to 13 (F4), and in Denmark from 0 (D7) to 11 (D3) (Supplementary Material S10). Total ant abundance in pitfalls was unimodal along both gradients, being greatest in the low open habitats with a woody overstorey of up to 50% cover (Supplementary Material S10). Site species richness was also strongly unimodal in Denmark and weakly unimodal in France, with highest values at intermediate latitudes (920 m) and vegetation cover (55%).

### 3.2. Geographic Patterns

Patterns of distribution along the two gradients varied greatly among genera (Figure 1). As predicted, the Cold-climate Specialists *Lasius* and *Formica* (*sanguinea*, *rufa* and *exsecta* species groups) occurred primarily at high latitudes and at high elevations at lower latitudes; Generalised Myrmicinae (*Crematogaster* and *Pheidole*) were found only at low latitudes and elevations; and Opportunists (*Myrmica*, *Tetramorium* and the *Formica fusca* species group) were widely distributed across the bioclimatic gradients.

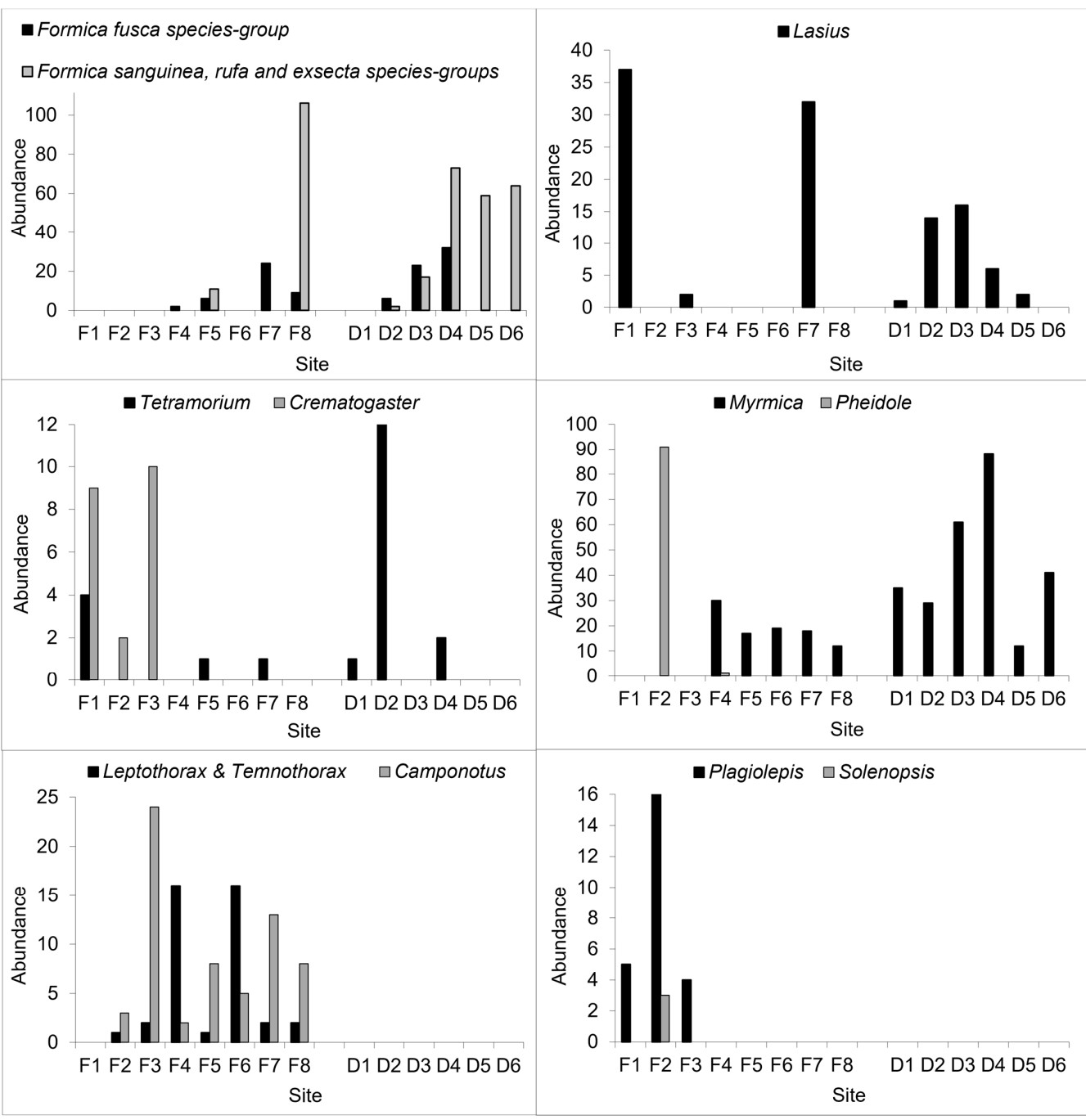

**Figure 1.** Distribution of major genera along the two bioclimatic gradients in Denmark (D1–D6) and France (F1–F8) of increasing thermal stress. Data are total abundance in pitfall traps. Site details are provided in Supplementary Materials S4 and S5.

Individual species had far more restricted distributions (Supplementary Material S7). For example, within *Lasius*, *L. emarginatus* occurred almost exclusively in the low-elevation open and dry French habitats, *L. fuliginosus* predominantly in the high-elevation French forests and open Denmark sites, and species of the *L. niger* species group exclusively in the Denmark sites. Similarly, *C. herculeanus* was restricted to the high-elevation French forests, whereas most other *Camponotus* species were collected only at the low-elevation French sites.

### 3.3. Behavioural Dominance

The most abundant species collected in traps at each site were also the most abundant species at baits (Supplementary Material S7). At French sites, with only one exception (F8), numerical dominance at baits was always greater than in pitfall traps, often markedly so (Supplementary Materials S7 and S8). In contrast, numerical dominance at baits was always less than in pitfall traps in Denmark. Total ant abundance at baits tended to increase with time, with maximum values at 60 min (Supplementary Material S11). A notable exception was site F2, where *Pheidole pallidula* was exceptionally abundant, and maximum abundance was attained after 30 min. At French sites, total ant abundance scores at baits after 60 min ranged from about 10 (mean of <1 ant per bait) to 60 (mean of about 15 ants per bait), with lowest scores occurring at mid elevation pine woodland sites. Total ant abundance at baits was consistently very low at Danish sites, with a maximum total score of only about 20 (mean of 2 ants per bait) after 60 min (Supplementary Material S11). There were no consistent patterns of ant abundance at baits along the two gradients.

Patterns of recruitment at baits (Supplementary Material S12) indicate that behavioural dominance was generally greatest at the low-latitude/low-elevation sites (F1–F3), as measured by a greater proportion of baits attended by ants and a greater number of high scores (>20 ants; scores 5–7) compared to other sites. Lowest behavioural dominance occurred at the Danish sites and the French high-altitude sites.

A range of species increased in abundance at baits over time, with the most common overall after 60 min being *Crematogaster scutellaris*, followed by species of *Myrmica*, *Camponotus*, *Lasius* and *Pheidole* (Figure 2). However, the frequency distributions of abundances scores (Figure 3) indicate generally low behavioural dominance. *Lasius emarginatus* had a right-skewed distribution, but it was recorded at only five baits (Figure 3A). *Crematogaster scutellaris*, *Pheidole pallidula* and *Lasius fuliginosus* were common species with relatively even abundance distributions, which is indicative of moderate behavioural dominance. The distributions of all other common species (*Lasius niger* and species of *Camponotus*, *Myrmica*, *Formica* and *Leptothorax*) were skewed to the left, indicating low behavioural dominance. Species of the *fusca* species group (Figure 3H) tended to have lower abundance scores than those from other species groups of *Formica* (Figure 3G), but no species of *Formica* was recorded with an abundance score greater than four (corresponding to a maximum of 20 ants).

### 3.4. Functional Group Classification

Distributional and behavioural results at baits and pitfall traps were as expected, so, with few exceptions, our functional group classifications of the European ant fauna (Supplementary Material S13) conform to those for North America (Supplementary Material S1). The major exception is that *Lasius flavus* and closely related species are removed from Cold-climate Specialists and now classified as Cryptic Species because they live an almost exclusively subterranean lifestyle. *Plagiolepis* was classified as a Cryptic species in [43], but we consider the European *P. pygmaea* to be an Opportunist.

Some European genera had not previously been classified into functional groups. We classified the social parasites and slave-makers *Rossomyrmex*, *Strongylognathus* and some *Tetramorium* as Cold-climate Specialists because their hosts (primarily species of *Formica*) have a predominantly cold-climate distribution. *Lepisiota* (not occurring in North America) is placed in Opportunists because of its widespread distribution and generalist habits. We

classified the *Formica* subgenera *Alloformica* and *Proformica* as Opportunists because of their distributional and behavioural similarity to species of the *F. fusca* species group.

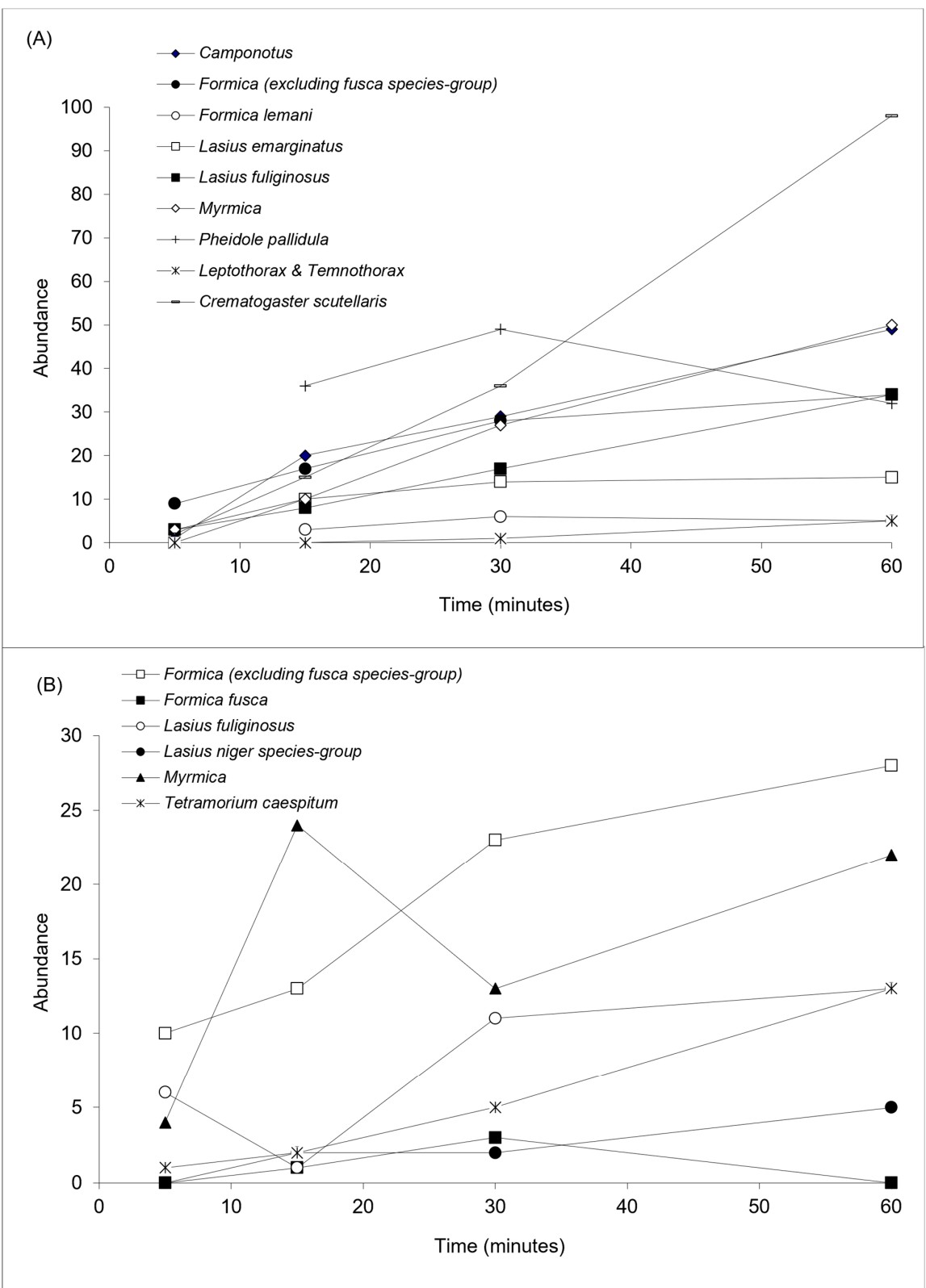

**Figure 2.** Temporal patterns of abundance at baits for major taxa for all sites combined in France (**A**) and Denmark (**B**).

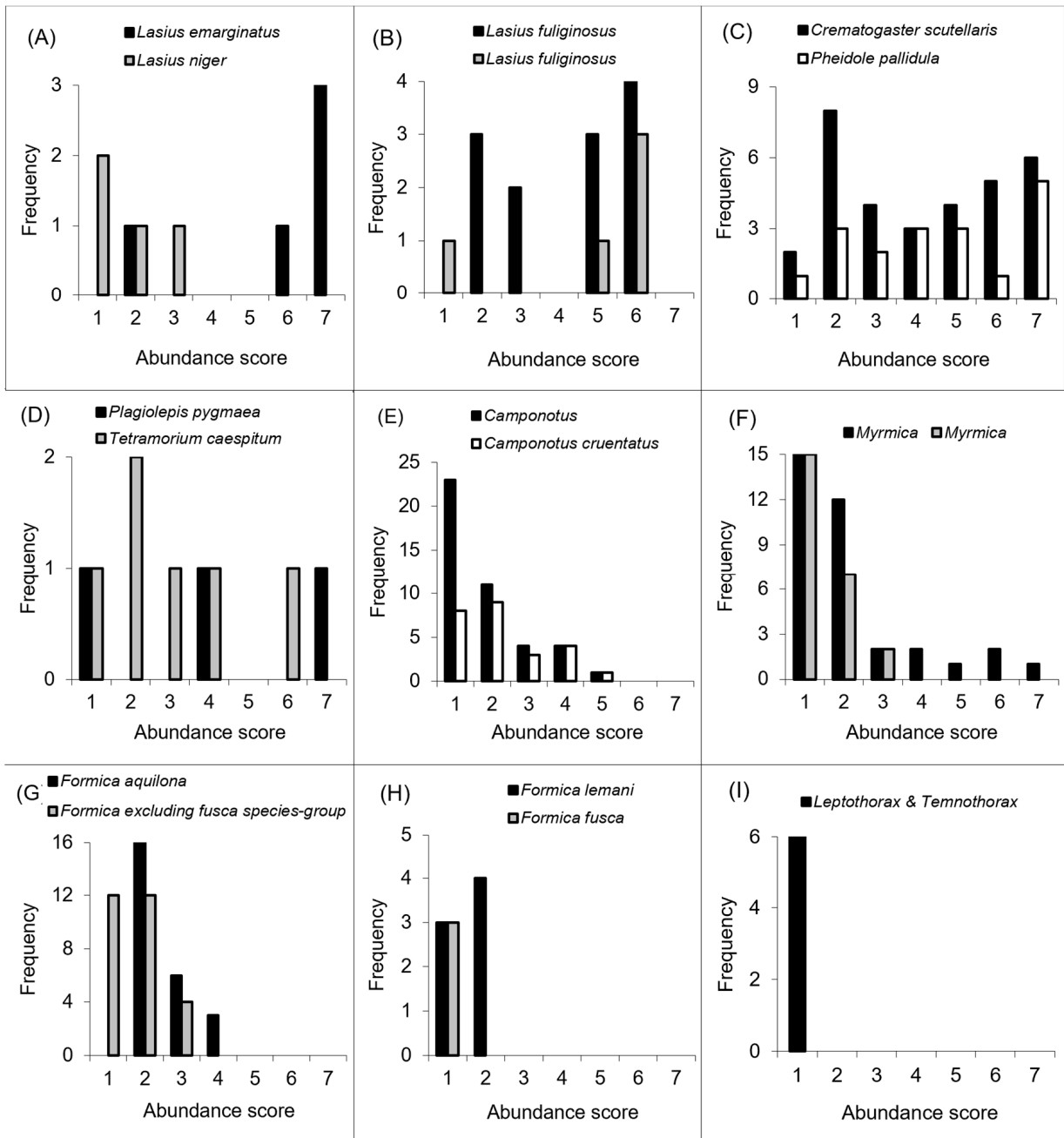

**Figure 3.** Frequency distributions of abundance scores at baits for all selected taxa (**A–I**); data pooled across sites along the vegetation gradients) in France (black or white) and Denmark (grey). Only 30 min and 60 min observation periods are included.

### 3.5. Biogeographic Patterns of Functional Group Composition

Functional group composition, as found by both pitfall traps and baits, varied systematically and predictably along both gradients (Figures 4–6). Generalised Myrmicinae (*P. pallidula* and *C. scutellaris*) occurred exclusively at warmer (low-elevation/low-latitude) sites, in contrast to the preference of Cold-climate Specialists for cooler sites (Figure 4). Opportunists tended to be most abundant at sites of moderate thermal stress derived from both expected climatic temperatures and levels of vegetation complexity. Subordinate Camponotini occurred throughout the French gradient (except for highest elevation site F8) but was not recorded at all in Denmark. Notably, the fauna of seasonally waterlogged

meadow site D1 in Denmark was comprised almost exclusively of Opportunists (species of *Myrmica*, *Tetramorium* and *Formica fusca* species group) (Figure 4).

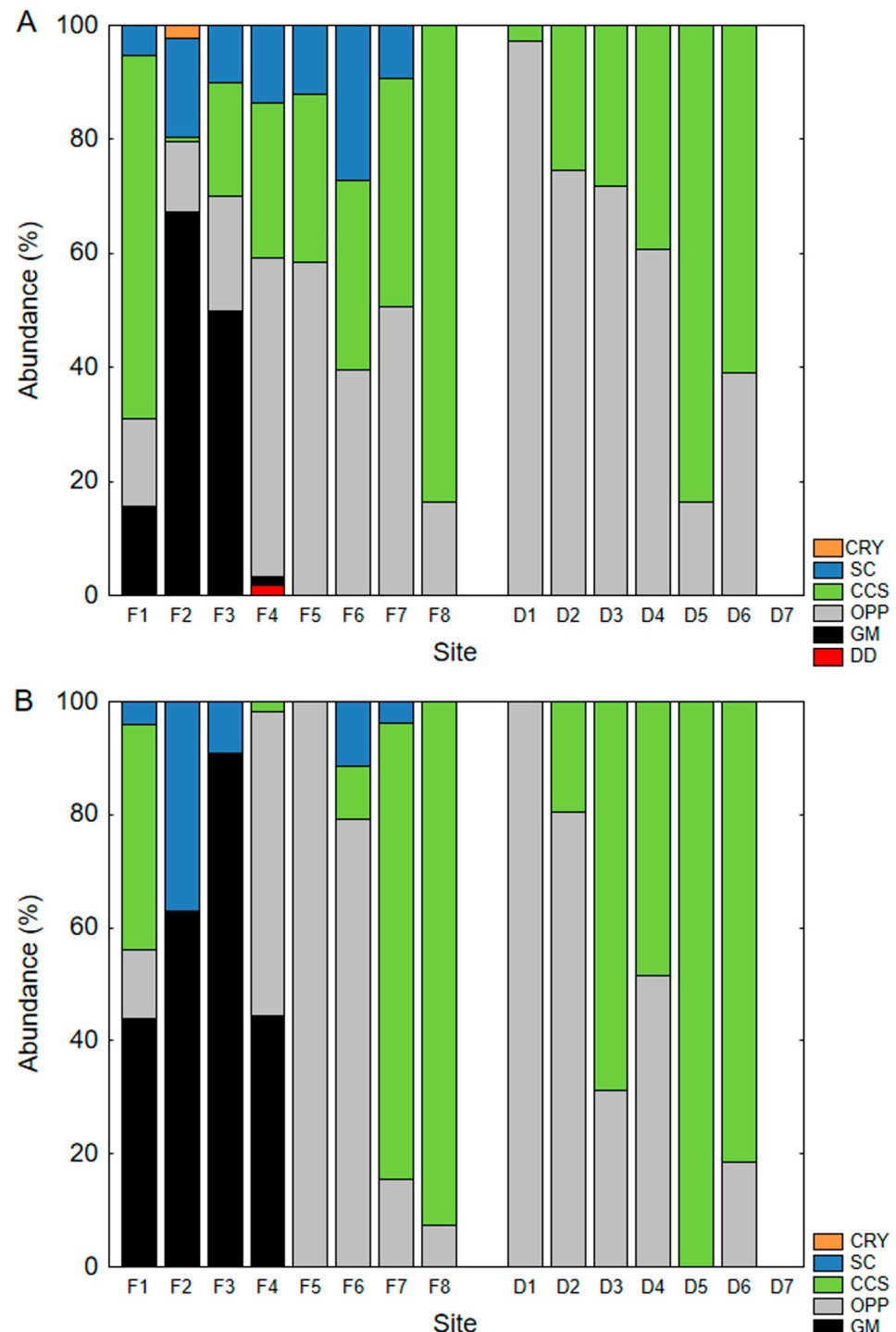

**Figure 4.** Functional group composition in pitfall traps (**A**) and at baits (**B**) at sites along the two bioclimatic gradients of increasing thermal stress in France (F1–F8) and Denmark (D1–D6). Bait data are for all times combined. The functional groups are: DD, Dominant Dolichoderinae; GM, Generalised Myrmicinae; OPP, Opportunists; CCS, Cold-climate Specialists; SC, Subordinate Camponotini; and CRY, Cryptic Species. Site details are provided in Supplementary Materials S4 and S5.

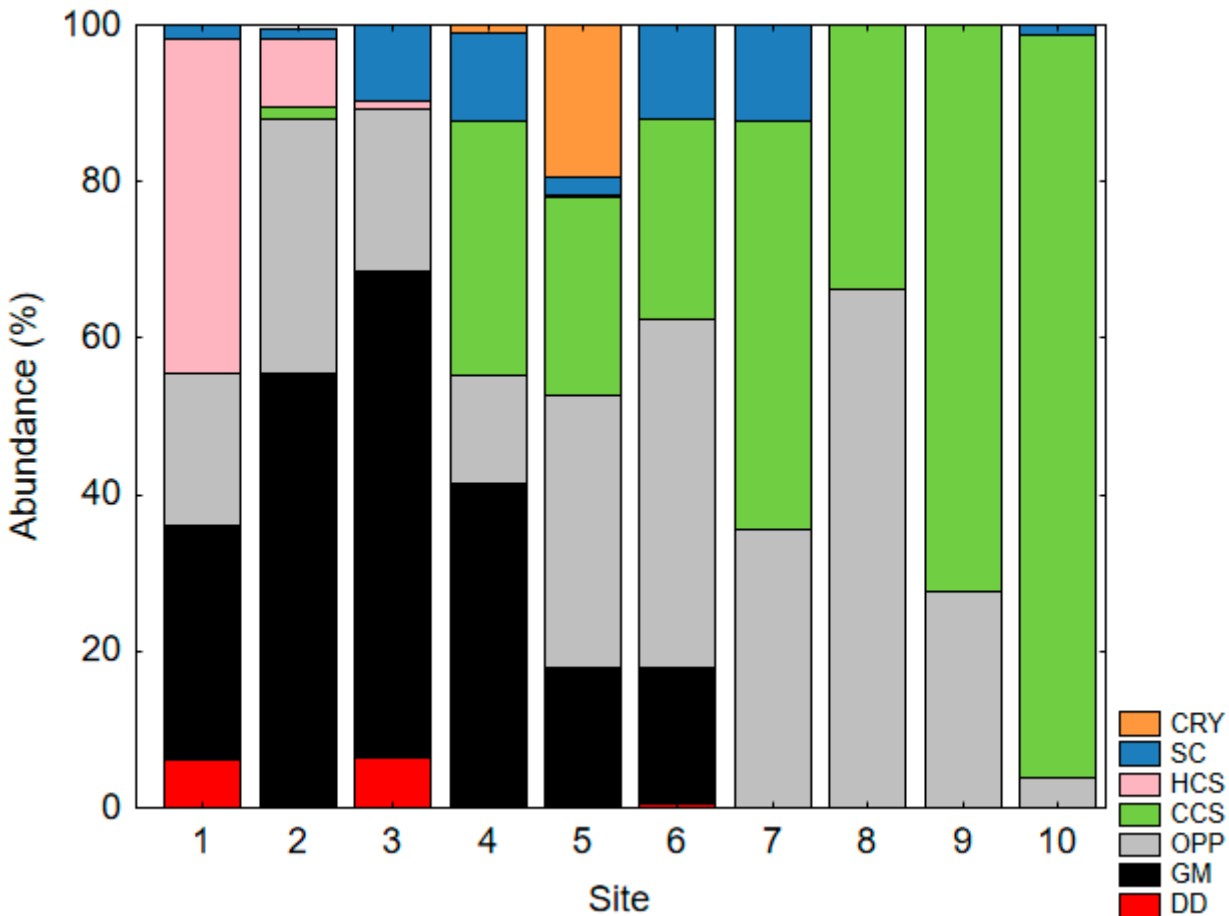

**Figure 5.** Functional group composition along a gradient of increasing thermal stress, combining results from the present study with published pitfall trap data. 1. grassland site in Spain [39]; 2. unburnt heathland site in south west France, summer unburnt 1992 sample [40]; 3. two Spanish woodland sites [39]; 4. shrubland sites F1 and F2 of the present study combined; 5. three Italian woodland sites combined [7]; 6. woodland sites F3–F6 of the present study combined; 7. forest sites F7–F10 of the present study combined; 8. woodland sites D2 and D3 of the present study combined; 9. forest sites D5 and D7 of the present study combined; 10. four Finnish forest sites combined (the Control sites of [41]). The functional groups are: DD, Dominant Dolichoderinae; SC, Subordinate Camponotini; HCS, Hot-climate Specialists; CCS, Cold-climate Specialists; CRY, Cryptic Species; OPP, Opportunists; GM, Generalised Myrmicinae.

The above biogeographic patterns were maintained when we combined our pitfall data with that from previous studies from throughout Europe, ranging from open grasslands in Spain to boreal forests in Finland (Figure 5). Dominant Dolichoderinae and Hot-climate Specialists occur exclusively in open habitats (grasslands and heathlands) of the Mediterranean region. Species of Generalised Myrmicinae occur primarily in such habitats, and they are increasingly replaced by Cold-climate Specialists with increasing thermal stress. Opportunists are well-distributed across the gradient, tending to be most abundant at moderate levels of thermal stress. NMDS revealed systematic compositional variation along the environmental gradient, with the sites ordering almost uniformly from right to left (Figure 6).

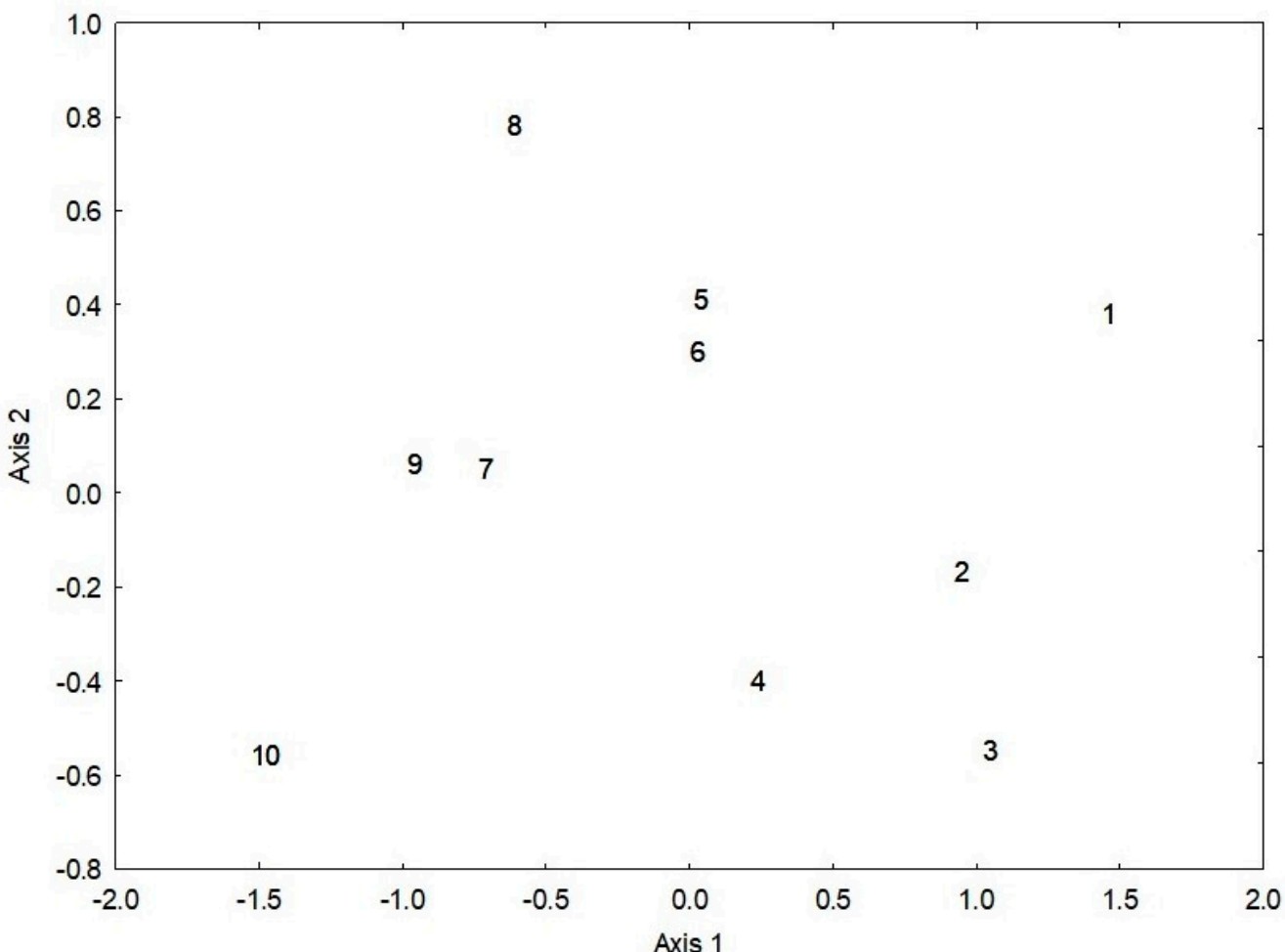

**Figure 6.** NMDS ordination of functional group composition of sites analysed in Figure 5 (points 1–10) representing an environmental gradient from a Mediterranean grassland (1) to a forest at high latitude (10). 2D Stress = 0.02.

## 4. Discussion

Our study applied a functional-group approach to investigate patterns of ant community composition in Europe in order to assess if patterns previously documented for North America apply elsewhere in the Holarctic. We first predicted that species richness and overall levels of behavioural dominance would decline with increasing thermal stress. We actually found a unimodal pattern of species richness at both the French and Danish gradients, with richness initially increasing but then decreasing markedly with increasing thermal stress and vegetation complexity. We recorded no ants at the most thermally stressed site (Beech forest) in Denmark. The decline in ant richness with increasing thermal stress is consistent with latitudinal and elevational patterns of ant diversity in Europe [7,19,22–24,44,45] and with broader ant diversity patterns globally [32]. The initial increase in richness along the elevation gradient in France can be attributed to an increase in habitat structural complexity, which has been well-documented as a key factor influencing ant species richness as well as species composition [20,25,46,47]. Overall levels of behavioural dominance as measured by abundance patterns at baits also declined systematically with increasing thermal stress, which is also consistent with patterns elsewhere in the world [4].

Our second prediction was that geographic patterns of key taxa conform to those in North America. As predicted, species classified as Cold-climate Specialists (most commonly from *Formica* (*rufa* species group) and *Leptothorax*) occurred primarily at high latitudes or

high elevations at lower latitudes. Generalised Myrmicinae (*Crematogaster* and *Pheidole*) showed the reverse pattern, occurring primarily at low elevations and latitudes Opportunists (species of *Myrmica*, *Tetramorium* and the *Formica fusca* species group) were widely distributed across bioclimatic gradients.

Our third prediction was that variation in behavioural dominance among taxa conforms to that in North America. This proved true in that the Generalised Myrmicines *Pheidole* and *Crematogaster* displayed moderate levels of dominance, as is the case globally [4,5], and other common genera such as *Tetramorium*, *Leptothorax* and *Temnothorax* displayed low behavioural dominance. In North America, the aggressive mound-building species of *Formica* (*rufa*, *sanguinea* and *exsecta* species groups) are behaviourally differentiated from other *Formica* species (particularly from the *fusca* species group), which are behaviourally submissive [5]. This was also the case in Europe, with the *fusca* species group showing poor discovery of, recruitment to, and defence of baits compared with other *Formica* species. However, we were surprised by the lack of high behavioural dominance exhibited by species of the *Formica rufa* group, given that they are widely regarded as the leading dominant species in Europe [12,48,49]. Such species (especially *F. aquilonia* in France and *F. rufa* in Denmark) were common at many of our sites but did not exhibit highly dominant behaviour at baits, having generally low abundance scores with left-skewed frequencies. This might reflect a relative lack of attractiveness to this taxon of the protein-based baits that we used, and the use of liquid-carbohydrate baits may have revealed a different pattern.

We did not record the behaviourally dominant dolichoderine *Liometopum*. We also collected only a single specimen of the *Tapinoma nigerrimum* species group (*T. erraticum*), which has been reported as behaviourally dominant in previous studies [7,16,28,31,39,50]. We classify both as Dominant Dolichoderinae and note that they have very restricted distributions in Europe.

From a global perspective, levels of behavioural dominance in the European ant fauna appear to be very low, even compared with North America [5] and certainly compared with Australia, where highly aggressive species of *Iridomyrmex* and other dolichoderines are ubiquitously dominant [4]. This is despite many European species from a range of functional groups being described in the literature as dominant, including species of *Formica*, *Lasius*, *Crematogaster*, *Pheidole*, *Camponotus* and *Tetramorium* [12–14,18,21,39,51]. However, behavioural dominance is a relative term, and species that are dominant over others locally are not necessarily dominant from a broader perspective [52].

## 5. Conclusions

For all common taxa, we found that patterns of distribution and behavioural dominance were consistent with their functional group classifications in North America. Such consistency is not surprising given the biogeographic similarity and shared evolutionary history of Europe and North America [51] and points to a functionally coherent ant fauna across the Holarctic. Our study illustrates how a functional-group approach can provide valuable insights into patterns of community organisation at biogeographic scales.

**Supplementary Materials:** The following supporting information can be downloaded at: https://www.mdpi.com/article/10.3390/d15030341/s1 Material S1: Summary descriptions of ant functional groups and associated North American taxa, updated from Andersen; Material S2: Map showing the locations of the study sites in Denmark (sites D1-D7) and France (sites F1–F8). The yellow scale bars are 500m; Material S3: Photographs of the study sites. Note that a photograph of site F4 is lost; Material S4: Summary habitat descriptions of the 15 sites distributed along environmental gradients in France (ordered according to elevation) and Denmark (ordered according to cover of woody vegetation), both gradients representing increasing thermal stress for ants; Material S5: Vegetation and ground-layer characteristics of sites along gradients of vegetation complexity in Denmark (D site codes) and France (F site codes). The height and cover of shrubs and trees are visual estimates, whereas all other data are means of 15 × 11 m quadrat samples; Material S6: Map of ten locations where data were sourced for Figure 5; Material S7: Ant species found at each study site in pitfall traps (p) at baits (b); Material S8: Species rank-abundance curves of ants quantified in pitfall traps (A–C)

and at baits (D–F) at sites in France (A, B, D, E) and Denmark (C, F); Material S9: Rarefaction curves of species accumulation at each site in France (A) and Denmark (B) as found in 15 pitfall traps and at 15 baits assessed after 5, 15, 30 and 60 minutes; Material S10: Ant abundance (pitfall trap catches only; black bars) and species richness (all records combined; grey bars) at sites along two bio-climatic gradients of increasing thermal stress in France (F1–F8) and Denmark (D1–D7); Material S11: Ant abundance at baits at each site along the two bio-climatic gradients in France (F1–F8) and Denmark (D1–D6) of increasing thermal stress for four observation times; Material S12: Frequency distributions of ant abundance scores at baits for each site along the vegetation gradients in France (A–H) and Denmark (I–N); Material S13: Assignment of European ant genera to functional groups.

**Author Contributions:** B.D.H. and A.N.A. designed the experiment. B.D.H. conducted the field work and analyses. B.D.H. and A.N.A. co-wrote the paper. All authors have read and agreed to the published version of the manuscript.

**Funding:** This research was funded by a CSIRO John Phillip Award to B.D.H.

**Institutional Review Board Statement:** Not applicable.

**Data Availability Statement:** All data are provided in the Supplementary Materials.

**Acknowledgments:** We thank Mogens Nielsen and Dorthe Birkmose in Denmark, as well as Jerome Orgeas in France for their hospitality. Thanks also to Perrine Poher for field assistance in France, and staff of the Mols Research Station in Denmark for allowing the work to be conducted there. Rhian Guillam and Xavier Espadaler provided valuable assistance with species identifications and Xim Cerda and multiple anonymous reviewers provided valuable comments on the draft manuscript. We are also very grateful to the researchers who supplied their ant community data. Ethical review and approval were not applicable for this study.

**Conflicts of Interest:** The authors have no conflict of interest.

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
