# Peer review of "Patterns of European Ant Communities Reveal a Functionally Coherent Holarctic Fauna"

_diversity, doi:10.3390/d15030341_

Round 1

Author Response

We thank the reviewer for their comments and suggestions. Please find following our responses in blue.

Reviewer 1

The sampling protocol (using pitfall traps as the main one) is adequate and combines three methods: pitfall traps, baits and hand collections. We are happy that the reviewer agrees that our sampling methodology is sound.

“Species rank-abundance curves from both pitfall catches and bait counts showed pronounced numerical dominance by a small number of species at each site (Supplementary Material 8).”: This would fit with a RAD curve that follows a geometric pattern. The authors should test the statistical fit to this model to enrich their claims. And usually, the Y axis has a logarithm scale. The fact that communities are dominated numerically by a small number of species is clearly shown by the graphs. We do not believe that formal statistical analysis is required to demonstrate this.

Line 319: “vegetation complexity” -> All the arguments in the paper are descriptive. At no time has the vegetation complexity been quantified. And therefore, no statistical study has been done. Delete or put as "possible / probable"  We have supplied quantified environmental measures in Supplementary Material 5. Note that we only use the term “vegetation complexity” in two places, lines 353 and 377. – we are unsure why the reviewer refers to line 319.

Totally agree with the last paragraph of the Discussion! We are very happy that the reviewer agrees with our last paragraph

Supplementary Material 7:

Replace Leptothorax lichtensteini by Temnothorax lichtensteini. Done, thank you for pointing this out

Replace Leptothorax nylanderi by Temnothorax nylanderi. Done, thank you for pointing this out

Supplementary Material 10:

Gradient: ordinal variable from 1 to 7 / 8 (depending on the area). It is possible to use Spearman's correlation coefficient (rS) to support the argument, with ants’ abundance and richness. And even the geographical distance of separation between (even the altitude) the stations can be used.  Our data show unimodal relationships and it is unclear what value Spearman correlation would provide. Similarly, it is unclear how the inclusion of geographic distance would add value when this is not an aim of our paper, especially with the Denmark sites which cannot be given a real gradient of distance or elevation.

Figure 6. NMDS ordination of functional group composition of sites:

The authors can calculate the matrix of geographical distances between the points.  With this matrix and the Bray-Curtis similarity matrix, it is possible to perform a Mantel test. The Mantel test (Mantel, 1967) may be used to calculate correlations between corresponding positions of two (dis)similarity or distance matrices. The Mantel test is widely used in biology to detect spatial structures in data or control for spatial correlation in the relationship between two data sets, for example community composition and environment. Alternatively, authors can use a Procrustes Analysis (least-squares orthogonal mapping) another method of comparing two sets of data. We note that some of this text comes from the GUSTA ME website, and that the website also clearly states numerous warnings of when not to use this test, especially for non-linear relationships, which is what we believe we have here. Additional text from the comment comes from the first sentence of the Abstract of the Legendre et al (2015) paper titled Should the Mantel test be used in spatial analysis, and the very next sentence says “We study demonstrates that this is an incorrect use of that test”. Even if such a test was appropriate, and such a test included generating a matrix of the environmental data, only half of the points on the graph are from our study, and very few of the other studies have provided the environmental data. We agree that if we were looking for patterns such as clustering of sites then we should and would conduct a statistical test (eg ANOSIM), but we don’t agree that a statistical test is needed to show the pattern that the sites are positioned in order along the primary axis from right to left.   

And finally, all conclusions have been “made by eye”, by guesswork, without the support of statistics. Fix as much as possible. And hopefully they are significant results!  As we state in the paper, our analytical approach follows exactly that of Andersen (1997).  We agree that many of our findings are ‘made by eye’ but do not agree that this equates to ‘guess work’. For example, formal statistical analysis is not required to show that Generalised Myrmicinae occurred exclusively at warmer (low-elevation/low-latitude) sites, in contrast to the preference of Cold-climate Specialists for cooler sites (Figure 4). The reviewer has not identified any patterns that they consider to be in doubt and therefore needs formal statistical support.

Reviewer 2 Report

The sampling protocol (using pitfall traps as the main one) is adequate and combines three methods: pitfall traps, baits and hand collections.

“Species rank-abundance curves from both pitfall catches and bait counts showed pronounced numerical dominance by a small number of species at each site (Supplementary Material 8).”: This would fit with a RAD curve that follows a geometric pattern. The authors should test the statistical fit to this model to enrich their claims. And usually, the Y axis has a logarithm scale.

Line 319: “vegetation complexity” -> All the arguments in the paper are descriptive. At no time has the vegetation complexity been quantified. And therefore, no statistical study has been done. Delete or put as "possible / probable"

Totally agree with the last paragraph of the Discussion!

Supplementary Material 7:

Replace Leptothorax lichtensteini by Temnothorax lichtensteini.

Replace Leptothorax nylanderi by Temnothorax nylanderi.

Supplementary Material 10:

Gradient: ordinal variable from 1 to 7 / 8 (depending on the area). It is possible to use Spearman's correlation coefficient (rS) to support the argument, with ants’ abundance and richness. And even the geographical distance of separation between (even the altitude) the stations can be used.

Figure 6. NMDS ordination of functional group composition of sites:

The authors can calculate the matrix of geographical distances between the points.  With this matrix and the Bray-Curtis similarity matrix, it is possible to perform a Mantel test. The Mantel test (Mantel, 1967) may be used to calculate correlations between corresponding positions of two (dis)similarity or distance matrices. The Mantel test is widely used in biology to detect spatial structures in data or control for spatial correlation in the relationship between two data sets, for example community composition and environment. Alternatively, authors can use a Procrustes Analysis (least-squares orthogonal mapping) another method of comparing two sets of data.

And finally, all conclusions have been “made by eye”, by guesswork, without the support of statistics. Fix as much as possible. And hopefully they are significant results!

Author Response

We thank the Reviewer for their comments and suggestions. Our responses are written below in blue

The manuscript aims to assess the reliability of ants’ functional groups described for American communities also for European communities. The authors used personally collected data and databases extracted from published studies to test whether functional groups are coherent between the two regions and if they actually describe the ecological functionality of the taxa along two different gradients, elevation, and tree coverage. They found that the functional groups describe for America are overall reliable even for European species, even pointing out belonging to a different functional group for some species. I found the manuscript very well-written, overall clear, and centered. Such a paper missed so far; the use of functional groups to analyze the ecological picture of the myrmecofauna in a region was sometimes debated and doubtful precisely because they were described for different biogeographic regions and biomes. This study fills this gap.  We are very happy to read all of these comments.

I did not find any particular problem to highlight. I only suggest the authors add a brief comment about the fact that data are not recent and the scenario might be different to date because of climate change, and that new samplings might be necessary in the future to confirm the results of the study. We acknowledge the importance of climate change species distributions. However, this would not affect the relationship between environmental stress and functional-group composition that is the basis of our paper. we describe. We therefore do not believe that such a caveat is required. 

In Fig. 3 there is a typo, the species is Formica aquilonia and not aquilona. Thank you for finding that typo.

Some figures appear to have a very low resolution, I suggest improving it. You may also use colors for symbols and bars in the graphs, it could be more clear and merely more pleasant for readers. The low resolution is simply the file size requirements for the paper submission. We have high resolution images to provide. We have also now changed some graphs to have colours rather than different grey-scale patterns.

In conclusion, I found the manuscript suitable for publication in its present form, except for the few suggestions above. We are very happy to read this positive assessment.

Reviewer 3 Report

The manuscript aims to assess the reliability of ants’ functional groups described for American communities also for European communities. The authors used personally collected data and databases extracted from published studies to test whether functional groups are coherent between the two regions and if they actually describe the ecological functionality of the taxa along two different gradients, elevation, and tree coverage. They found that the functional groups describe for America are overall reliable even for European species, even pointing out belonging to a different functional group for some species.

I found the manuscript very well-written, overall clear, and centered. Such a paper missed so far; the use of functional groups to analyze the ecological picture of the myrmecofauna in a region was sometimes debated and doubtful precisely because they were described for different biogeographic regions and biomes. This study fills this gap.

I did not find any particular problem to highlight. I only suggest the authors add a brief comment about the fact that data are not recent and the scenario might be different to date because of climate change, and that new samplings might be necessary in the future to confirm the results of the study.

In Fig. 3 there is a typo, the species is Formica aquilonia and not aquilona.

Some figures appear to have a very low resolution, I suggest improving it. You may also use colors for symbols and bars in the graphs, it could be more clear and merely more pleasant for readers.

In conclusion, I found the manuscript suitable for publication in its present form, except for the few suggestions above.

Author Response

We thank the Reviewer for their comments and suggestions. Our responses are below in blue

Title: Patterns of European ant communities reveal a functionally coherent Holarctic fauna

Authors examine whether functional composition of European ant assemblages conforms with those documented in North America, using field survey in France and Denmark and other studies. As explained in text, the functional group tested here is widely known to ant ecologists. Field survey and data analysis are properly performed, and results are also properly represented. I think that this paper is good for this journal. We are very happy to read this positive feedback

However, there are some questions, and a suggestion for data representation.

  1. Functional classification in Supplemental Material 13

To my knowledge in ant fauna in Asia which is also one of three main regions (Europe, NA, and Asia) in Holartic region, a few of generic classification are questionable to me. Polyergus samuraiin east Asia is slave maker rather than specialist predator. I assume that some European Polyergus species are also slave maker. Please check it. I do not agree that Temnothorax is cold climate specialist. Thermal range of this genus is wide. Some Temnothorax species inhabit in low altitudinal area with non-cold climate in my study region.

We have followed Andersen (1997) in the classification of slave-making Polyergus as Specialist Predators because they capture brood of other ants (albeit do not eat them). Species are classified as Cold-climate specialists based on the species-group to which they belong more generally. Yes, some species of Temnothorax do occur in warmer climates but the genus as a whole is overwhelmingly distributed in higher latitudes/altitudes.

  1. Behavioral dominance using abundance in baits.

I do not understand that abundance in baits represent behavioral dominance. In addition to behavioral dominance, abundance in baits may be determined by other factors such as abundance (population size), presence of other species, attraction to baits, and foraging activity. Low abundance in baits in Denmark may be due to low foraging activity of ants rather than due to low behavioral dominance. Ants in low temperature area such as Denmark may have lower foraging activity compared with ants in high temperature area such as France. This phenomenon appears even in similar temperature during survey. I do not understand why abundance score in Fig. 3 is related with behavioral dominance of each species. To my experience, behavioral dominance is clearly revealed by interaction (behavior) between species. Dominant species shows aggressive behaviors such as attack toward submissive species. Yes, we agree completely with all the reviewer says here about dominant species. From the perspective of the functional group scheme, species that are truly behaviourally dominant at the global scale (not the local scale), will display all of these traits consistently. For figure 3, in this part of the measure of “Dominance”, a truly globally dominant species would have a right-hand skewed graph (ie consistently having high abundance at baits whenever present because it has high abundance, because it can recruit, and because it is aggressive and can either take a bait from another species and/or can defend that bait). The graphs is figure 3 show that none of the species have a right-hand skewed distribution, and are either left-hand skewed, or something in between. One thing that we state in the paper that is demonstrated here, is that, at the global scale, there are no truly globally dominant ant species at the higher latitudes – such species live in warmer climates. Dominance is relative.